# Anti-Cocaine IgA Rather Than IgG Mediates Vaccine Protection from Cocaine Use

**DOI:** 10.3390/pharmaceutics14112368

**Published:** 2022-11-03

**Authors:** Thomas R. Kosten, Colin N. Haile, Coreen B. Domingo, Elizabeth B. Norton

**Affiliations:** 1Menninger Department of Psychiatry, Baylor College of Medicine, Houston, TX 77030, USA; 2The Michael E DeBakey Veteran’s Affairs Medical Center, Houston, TX 77030, USA; 3Department of Psychology & TIMES, University of Houston, Houston, TX 77204, USA; 4Department of Microbiology & Immunology, Tulane University School of Medicine, New Orleans, LA 70112, USA

**Keywords:** vaccine, cocaine, clinical trial, IgA, IgG

## Abstract

In developing a vaccine for fentanyl use disorder, we observed that IgA was the best correlate of vaccine-mediated protection from injected drug challenge, rather than IgG or binding affinity. Recent evidence shows that IgA secreting cells line the blood–brain barrier that capture pathogens and could prevent drug antigens from penetrating the brain. We assayed IgA and IgG antibodies from an anti-cocaine vaccine clinical trial and categorized each subject’s antibody levels using half-log cut-points for IgA: <1000, <5000, <10,000 and >10,000; and for IgG: <10,000 to >100,000. We compared these antibody groups on urine toxicology in 130 subjects at week 9 after 3 booster vaccinations. We also provided relevant data on benzoylecgonine (BE, cocaine metabolite) from this study’s placebo patients. BE urine levels were lowest for the highest IgA category; however, levels did not differ across IgG groups. Our findings linking IgA to protection from cocaine and fentanyl in mice, rats and humans are novel and suggest an increasingly recognized role of IgA in vaccine efficacy.

## 1. Introduction

Substance use disorders (SUD) are a major health issue that cause personal and societal suffering and associated economic losses [1]. Relapse rates are high with cocaine (COC) SUDs, and recently COC has joined fentanyl (FEN) as a major cause of epidemic lethal overdoses [2]. No pharmacotherapy is FDA approved for COC SUD, but an effective vaccine therapy could produce antibodies (AB) to block COC’s physiological, behavioral and reinforcing effects by preventing the COC from penetrating the brain [3]. Anti-COC AB induced by a vaccine could prevent relapse after attaining abstinence and reduce the recent epidemic of COC overdose deaths [2].

Vaccines against SUDs have been developed since the 1970’s [3], and some vaccines against nicotine and cocaine have moved to clinical trials, but with limited success [4,5,6]. Many factors contributed to those failures including using a traditional combination with alum adjuvant and a potentially misdirected focus on enhancing immunoglobulin G (IgG) titers rather than other types of AB. The failures of these previous vaccines were thought to be based on poor achievement of high titer serum IgG with anti-cocaine or anti-nicotine vaccines [4,5,6,7].

High titer serum IgG was recently shown after vaccination of rodents with a fentanyl (FEN) hapten conjugated to a CRM197, a carrier protein admixed with the adjuvant dmLT (double-mutant heat-labile toxin) or LTA1 (heat-labile enterotoxin from *E. coli*) [8]. These high AB levels also markedly blocked FEN-induced analgesia and FEN penetration into the brain compared with alum-adjuvanted vaccination in mice [8]. A strong component of this efficacy was using an LT-based adjuvant, which is derived from heat-labile enterotoxin [9,10]. The dmLT derivative has been successful in several Phase I/II clinical trials for microbial vaccines [11,12,13,14,15,16,17]. This success is a critical advance since no single adjuvant has emerged as ideal for SUD vaccines although alum, a Toll-like receptor-based agonist (TLR) or combinations thereof, have been attempted [18].

Perhaps more importantly, we observed that immunoglobulin A (IgA) was the best correlate of vaccine-mediated protection from an injected drug challenge, rather than IgG, IgG1, IgG2a or binding affinity [8]. We recently replicated this IgA association with blocking analgesia and reducing brain levels of FEN in rats [19]. IgA is typically thought to prevent mucosal microbial or drug exposures, and our findings about IgA for blocking injected drugs is a critical discovery. These findings are consistent with new work showing that IgA cells line the blood–brain barrier and specifically make IgA that could exclude antigens like FEN and COC from the brain [20]. In the current study we re-examined samples from an anti-COC vaccine clinical trial and found support for IgA’s role in drug protection based on urine toxicology results [7].

This support for the role of IgA from a human anti-COC study is exciting because of the cross-species replication, but also because this human study used only alum as an adjuvant [7]. Thus, producing sufficient IgA levels to show this association with the key behavioral outcome of reducing continued COC use based on urine COC metabolite levels was not dependent on this new adjuvant of dmLT.

## 2. Materials and Methods

### 2.1. Study Design

The blood samples for this study were provided from this 6-site, 24-week, Phase III randomized double-blind placebo-controlled trial assessing succinyl-norcocaine conjugated to cholera toxin B protein and using an alum adjuvant as a vaccine for cocaine use disorder [7]. We measured urine cocaine metabolites (BE) thrice weekly as the main outcome, and 130 of 150 initially active vaccine subjects remained in treatment through week 9, when the series of four vaccinations (1, 3, 5, and 9 weeks) were completed. For the current study, we compared these week 9 urine results to the anti-COC AB titers at week 9. We also used the BE values at baseline before antibodies would form from vaccination in the actively vaccinated and BE values at baseline and week 9 from the 116 placebo subjects who remained through week 9. Based on previous studies [6,21] we had a target anti-COC IgG level of ≥42 μg/mL (high IgG) as a minimal level needed for efficacy, and 67% of the 130 vaccine subjects attained that level. In total, 76% of the subjects were male and 72% were African American, with a mean age of 46 years, and had smoked cocaine on average for 13 days per month at baseline. Written informed consent was through Baylor College of Medicine via protocol H-23257 (Multisite Controlled Trial of Cocaine Vaccine TA-CD).

### 2.2. Quantitative Antibody Measurement

We stored all plasma samples at −80 °C in individual small glass vials which we unfroze for sample aliquots no more than three times during their 8 years of storage. Serum anti-cocaine IgG and IgA were measured by an Enzyme-Linked ImmunoSorbent Assay (ELISA). Briefly, cocaine conjugated to bovine serum albumin (BSA) was bound to ELISA plates overnight (2 ug/well) and then incubated with dilutions of patient serum. After washing, human IgG or IgA antibodies bound to the plates were detected with anti-human IgG or IgA antibody conjugated with horseradish peroxidase and an appropriate substrate [6]. Background antibody binding to the target conjugate protein (BSA) alone was subtracted, and each ELISA plate included wells with serially diluted polyclonal human IgG and IgA to provide an internal standard curve. Absorbance was measured at 450 nm using a plate reader (Multiskan Go, Thermo Scientific, Waltham, MA, USA). Third-order polynomial curve fitting of cocaine standard comparators was used to quantify anti-cocaine IgG and IgA concentrations. This ELISA’s specificity and reproducibility was validated using serial dilutions of a humanized monoclonal IgG antibody to cocaine, gA2E2, from Dr. Andrew Norman at the University of Cincinnati [22,23].

### 2.3. Urine Cocaine Metabolites

Urine was qualitatively tested for benzoylecgonine (BE) at the time of screening and thrice weekly for the duration of the 16-week study and 8-week follow-up. Urine BE values of at least 300 ng/mL were considered positive. Pharmacokinetic studies have shown that anti-COC AB in the circulation did not change BE excretion, and our subjects’ anti-COC IgG and IgA had less than 15% cross-reactivity with BE [6,24].

### 2.4. Data Analyses

While both urinary BE and serum IgG and IgA are all continuous measures, they were quite non-normal in their distributions with skewness of 3.2, 1.4 and 1.9 and with kurtosis of 14.5, 2.2 and 4.3, respectively. On graphic display, the values of all three assessments showed twin peaks with the secondary peaks at substantially greater values than the main peak. To address these non-normal distributions of values we used binning based on half-log values for the immunoglobin values and applied Kruskal–Wallis non-parametric statistics for comparing those bins due to the substantially skewed (3.2) and kurtotic (14.5) urine levels (Graph Pad Prism v. 9, using *p* ≤ 0.05). Because the immunoglobin values were more than 10-fold larger for IgG than IgA, the bin value distributions substantially differed between IgG and IgA. The IgA group had a sufficiently wide distribution of IgA levels (kurtosis 4.3) to allow four bins containing 5 to 88 observations per bin, whereas the IgG group had a tighter distribution of IgG levels (kurtosis 2.2) allowing only three bins containing 8 to 80 observations per bin, as shown in Figure 1. The figure shows these categories ranging from <49,999 to >100,000 ng/mL for IgG and from <1000 to >10,000 for IgA. We examined the urinary BE levels (ng/mL) obtained during week 9 for each subject.

## 3. Results

### IgG and IgA Antibody Levels

The two bar graphs in Figure 1 present the association between the urinary COC (BE) levels in ng/mL to the IgG and IgA concentrations during week 9, when anti-COC AB levels were optimal in the subjects. This comparison shows categories’ standard errors of the mean and individual values for each subject. We performed nonparametric Kruskal–Wallis tests controlling for multiple testing across the four IgA groups. Visual comparison and statistical analyses indicated no statistically significant differences in urinary BE among the IgG concentration groups. However, a significant effect was observed and statistically confirmed across the stair-step decline in BE levels for the four IgA groups between <1000 and 10,000+ IgA levels (*p* < 0.05). Simple correlations between the urine BE and IgA showed no association due to the relatively small number of patients with high IgA levels suggesting the need for a more potent vaccine. We did not find that the combination of IgA and IgG binding showed a stronger association in reducing the urine BE levels.

## 4. Discussion

This analysis showed significantly reduced COC use with the relatively high IgA titers above 10,000 ng/mL, and a stairstep reduction in COC use as the IgA titer rose through the three categories of: (a) <1000, (b) >1000 and <10,000, (c) >10,000 ng/mL. The IgG titer did not show such an association with COC use. Targeting both IgA and IgG for SUD vaccines is paradigm shifting, although this study did not find an additive effect for these two immunoglobins in reducing urine BE levels. Instead, these outpatient subjects, who had relatively free access to cocaine during this clinical trial, made apparent attempts at over-riding the active vaccine blockade, as observed in Figure 1. This Figure shows very high individual values for BE in the patients with relatively lower IgG or IgA levels (e.g., BE = 700,000 and 400,000 for IgG and IgA, respectively). To further show that the actively vaccinated subjects increased their cocaine intake after attaining effective antibody levels, we can compare urine BE levels from this study’s placebo patients. At week 9 the maximum urinary BE level was 763,000 ng/mL for the placebo patients compared to 1,031,000 ng/mL for the actively vaccinated. This 34% greater BE level for the vaccinated at week 9 indicated more cocaine use than in the placebo subjects. Furthermore, at week 1, before any antibodies could form, the mean urine BE level was 15% lower for those who received active vs. placebo vaccine (e.g., 74,000 ng/mL active vs. 88,000 ng/mL placebo). This attempt to over-ride the vaccine blockade can be stated alternatively; the active vaccine patients rose 13-fold from their average baseline of 74,000 ng/mL to their maximum BE levels at week 9, whereas the placebo patients rose only 7-fold from baseline. Since the placebo subjects had no blockade to over-ride, they showed about half of the increase in their cocaine use compared to the actively vaccinated subjects.

IgA antibodies have been reported after COC and nicotine intranasal vaccines that protected animals from corresponding drug challenges. IgA has also long been appreciated for high affinity binding to toxins and microbes at mucosal tissue, where most IgA secreting plasma cells reside. IgA is also found in cerebral spinal fluid of healthy humans in both monomeric and dimeric forms [25]. A central question remains, is drug-specific IgA directly protective against COC or other drug parenteral or smoked challenge more than IgG subclasses or is this IgA a surrogate of other protective immune responses not yet defined? It may also be the combination of IgA and IgG that contributes to protection. We now believe that a novel IgA inducing multivalent vaccine is possible and would be the best option for using a SUD vaccine in human populations based on the previous rodent and this human study [8]. Future studies should evaluate the role of IgG and IgA isotypes in SUD vaccine outcomes.

Our findings linking IgA to protection from abused drugs in mice, rats and humans are completely novel and this is the first study assessing IgA with any anti-cocaine vaccine in humans. Our previous clinical trial from 2009 (6) found that anti-cocaine IgG levels above 40 mcg/mL significantly reduced BE urine levels in cocaine abusers. The current analysis of the larger 2014 study did not find any association between anti-cocaine IgG levels including using the previous therapeutic cut-off of 40 mcg/mL (7). However, an independent research group found that meningeal IgA is essential for defending the central nervous system from infectious organisms [20]. This group also identified that mucosal (e.g., oral) antigen exposure results in these protective IgA+ plasma cells migrating to the meningeal venous sinuses around the brain that help ‘capture’ threats to the brain, such as fungal organisms. We believe this may be a contributing mechanism at work with an effective vaccination approach. Thus, future studies might use depletion and imaging techniques to evaluate if anti-FEN and anti-COC IgA directly protects the brain from injected drug challenge.

## 5. Conclusions

In conclusion, we found that IgA, and not IgG levels, were associated with decreases in cocaine-positive urines in individuals that received our succinyl norcocaine cholera toxin (SNC-CTX) conjugate vaccine. This study expands our studies in rodents to humans and further highlights the potential role of IgA in vaccine efficacy for SUDs.

## Figures and Tables

**Figure 1 pharmaceutics-14-02368-f001:**
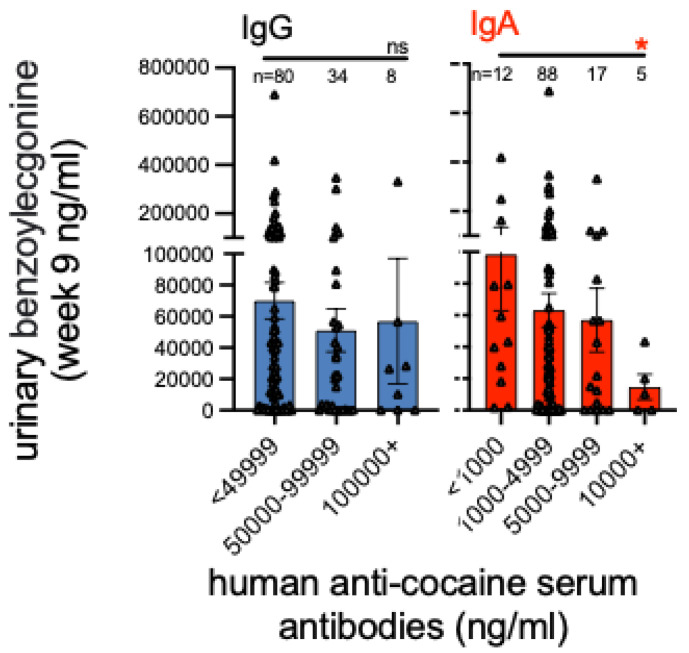
Anti-cocaine IgA is implicated in vaccine mediated protection from cocaine use in a human cocaine vaccine trial. Urine cocaine metabolites (benzoylecgonine) from a Phase III cocaine vaccine study were compared to serum anti-cocaine IgG and IgA levels at week 9 post-vaccination when peak antibody levels were found. Bars represents mean urinary cocaine levels (S.E.M.). ns = not significant, * *p* < 0.05.

## Data Availability

Not applicable.

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
