# Peer review of "Anti-Cocaine IgA Rather Than IgG Mediates Vaccine Protection from Cocaine Use"

_pharmaceutics, 2022, doi:10.3390/pharmaceutics14112368_

Round 1

Reviewer 1 Report

Overall Summary

This communication by Kosten et al. relates a reanalysis of samples collected from an anti-cocaine vaccination trial whose top-line results were previously reported in 2014, with the basis for this reanalysis being an assessment of whether IgA concentrations have a greater association with urine metabolite markers of cocaine use in comparison to IgG. This is an important translational question, given relevant associations with IgA function in rodent studies and improved understanding of IgAs' roles at the blood-brain barrier. While the findings are interesting, questions regarding the choice of analytical and statistical approaches used need to be addressed prior to publication.

Point-by-point remarks

Page 2 - Line 45 (P2-L45; Minor): Please provide the full descriptions for the acronyms dmLT and LTA1 upon first use

P2-L72 (Minor): Given the amount of time occurring between the sample collection and the re-analysis of these samples, providing information about sample storage conditions would be helpful to the reader.

P3 - L87 (Minor): Changing to "anti-cocaine IgG and IgA were measured by...", and then eliminating the sentence "A similar procedure was done..." on L99 would be helpful for clarity, given that IgA is the main focus of this paper

P3 - L107 (Middling): The cross reactivity of IgG with Benzylecgonine is noted, but not the cross-reactivity of IgA with benzylecgonine. Since IgA binding to BE could alter its PK profile, including excretion into the urine, this would likely be a useful factor to know.

P4-L121 (Major): Since both urinary BE concentration and serum concentration of IgG and IgA are continuous measures, it is unclear why binning the data is necessary for this analysis. Directly assessing correlations between BE and Ab concentration measures, at least in addition to the binned analysis, could provide more rigor. If there is a particular reason why this usual form of continuous-continuous data analysis cannot or should not be applied, additional discussion about this should be included by the authors. 

P4-L121 (Major): It is noted that non-parametric t-tests were used for the analysis between groups. However, even within the IgG and IgA binned comparisons, more than 2 groups are being compared. Presently, no information is included about correction for multiple comparisons, or why an ANOVA, or a nonparametric analogue of the same, is not being used. For claims about comparisons across IgG and IgA groups, a Two-Way ANOVA would seemingly be a rigorous addition allowing one to make claims about their relative relevance, as is being alluded to in P5-L134-137. As above, including a discussion about why these usual approaches were not taken would be helpful, at a minimum.

Display - Figure 1 (Minor): The y-axis label might be more easily interpreted by the reader if it read 'Urinary benzylecgonine', as that appears to be the measurement that was undertaken.

P5-L145 (Minor): With regard to whether the combination of IgA and IgG is more predictive of protection, commentary as to whether such a trend is observable in analysis of this data set would be helpful.

Author Response

Dear Reviewer # 1:

Please accept our thanks for your time and expertise through the review process. Your critique has provided us an opportunity to further strengthen our manuscript and better clarify salient details. Please see the attached document.  

Thank you, in advance, for your time in the re-review process. 

Sincerely,

Thomas R. Kosten, MD et al. 

Reviewer 2 Report

The manuscript represents a significant advancement in the understanding of efficacy of substance abuse vaccines. The correlation of IgA titer with the decrease in cocaine in urine in humans is a major advancement. 

The study execution is sound and the manuscript is well written. I have no questions or changes.

The manuscript should be accepted for publication.

Author Response

REVIEWER # 2

  1. The study execution is sound and the manuscript is well written. I have no questions or changes.
  • Response: Please accept our thanks for your time and expertise through the review process. We are grateful for your assessment of our manuscript. Thank you. 

Reviewer 3 Report

The present short communication deals with cocaine vaccination and deserves attention, since there is currently no real or efficient treatment against drugs of abuse. However, as recently reviewed, vaccines are not an option for treating nicotine or cocaine abuse and there is a need for novel treatment strategies (BT Bloom, 2022) except notably for substitute treatments for opioid dependence with subutex or methadone. The present communication with a single figure reports a decrease in urinary cocaine level (p<0.05) when using IgA vs IgG antibodies from an anti-cocaine vaccine. It falls short on many aspects in respect to the journal IF.

Major issues should be addressed and/or discussed

1. Benzoylecgonine (BE) and cocaine should be evaluated in blood samples.

2. Are the data of IgG and IgA consistent with previous studies including those of the authors?

3. Since antibodies prevent cocaine penetrating into the brain, whether patients with free access to the drug increase their intake should be addressed.

4. In the single figure, N numbers should be indicated for each anti-cocaine serum antibody concentration.

5. Within the abstract the conclusion for Substance use disorders is an overstatement based on limited provided data and in respect to other publications.

Minor concerns: abbreviations should be more clearly defined for potential readers.

Author Response

Dear Reviewer # 3:

Your time and expertise in reviewing our manuscript submission is most greatly appreciated. Attached is a point-by-point response to your helpful comments. Thank you for your re-review. 

Thomas Kosten, MD et al. 

Round 2

Reviewer 3 Report

The revised manuscript is fine